# A Sensitive and Specific Monoclonal Antibody Based Enzyme-Linked Immunosorbent Assay for the Rapid Detection of Pretilachlor in Grains and the Environment

**DOI:** 10.3390/foods13010012

**Published:** 2023-12-19

**Authors:** Linwei Zhang, Xiaoyang Yin, Hongfei Yang, Hao Wen, Shiyun Han, Xiaoming Pan, Huaming Li, Dapeng Peng

**Affiliations:** 1National Reference Laboratory of Veterinary Drug Residues (HZAU) and MOA Key Laboratory for Detection of Veterinary Drug Residues, Huazhong Agricultural University, Wuhan 430070, China; zhanglinwei9910@163.com (L.Z.); yinxy73251@163.com (X.Y.); tiep990212@163.com (H.Y.); w793653717@163.com (H.W.); hanshiyun2022@163.com (S.H.); pxiaoming2596@163.com (X.P.); lihm9915@163.com (H.L.); 2State Key Laboratory of Agricultural Microbiology, Huazhong Agricultural University, Wuhan 430070, China; 3Shenzhen Institute of Nutrition and Health, Huazhong Agricultural University, Wuhan 430070, China; 4Shenzhen Branch, Guangdong Laboratory for Lingnan Modern Agriculture, Genome Analysis Laboratory of the Ministry of Agriculture, Agricultural Genomics Institute at Shenzhen, Chinese Academy of Agricultural Sciences, Shenzhen 518000, China

**Keywords:** residue detection, pretilachlor, monoclonal antibody, ic-ELISA

## Abstract

Pretilachlor is a chloroacetamide herbicide mainly used for weed and broadleaf weed control in rice, that is widely utilized in China. In order to detect the residue of pretilachlor in the environment and food, a highly sensitive and specific monoclonal antibody (mAb) against pretilachlor was prepared, and the half maximum inhibitory concentration (IC_50_) of the monoclonal antibody was validated to be 31.47 ± 2.35 μg/L. An indirect competitive ELISA (ic-ELISA) based on the antibody with a linear range of 6.25~100 μg/L was developed. The specificity of the antibody was explained by computer simulations and experimental validation. The mAb exhibited negligible cross-reactivity towards alachlor, acetochlor, propisochlor, butachlor, and metalaxyl, and the limits of detection (LOD) for pretilachlor in lake, rice, and soil samples were 4.83~5.23 μg/L. The recoveries of all samples were 78.3~91.3%. The reliability of the ic-ELISA method for residue detection of pretilachlor in the environment and grains was confirmed using high performance liquid chromatography.

## 1. Introduction

In recent years, the development of chloroacetamide herbicides has been rapid, with annual production, application range and usage area second only to that of organophosphorus herbicides [1,2]. Pretilachlor (PR) is a highly effective, low-toxicity, broad-spectrum, post-sowing, pre-seedling selective chloroacetamide herbicide. Its exceptional water solubility and penetrability enable effective weed control, leading to enhanced crop yield. Consequently, it has gained widespread application in rice transplanting and weed management practices [3]. But its overuse has potential problems, such as increased weed resistance and contamination of the surrounding environment; PR has also shown toxic effects on fish [2,4,5,6,7], and algae [8,9]. It is important to note that the perceived low toxicity of PR is relative and prolonged ultra-limited exposure still carries risks. Some studies have shown PR to be neurotoxic, genotoxic and carcinogenic [10], its enrichment in the food chain ultimately harms human beings [11,12]. To control its contamination of the environment and food, clear residue limits for PR have been established in various countries. In China, the maximum residue limit (MRL) for rice is set at 0.1 mg/kg, while for wheat it is 0.05 mg/kg [13].

The most commonly used methods for residue analysis of PR in domestic and overseas are gas chromatography and liquid chromatography, Bai et al. developed a method for the determination of chloroacetamide herbicide residues in water samples by dispersive liquid-liquid microextraction and gas chromatography with flame ionization detection (GC-FID). Additionally, Nguyen Dang Giang Chau et al. established a method for determining PR residues in vegetables through gas chromatography-tandem mass spectrometry (GC-MS/MS). The limit of detection for both methods ranged from 0.01 to 10 μg/L [14,15]. Gao et al. established a high-performance liquid chromatography-tandem mass spectrometry (HPLC-MS/MS) method for the determination of PR in fish with a detection limit of 0.01~0.1 μg/kg [16]. These instrumental assays offer the advantage of high sensitivity and accuracy; however, the market is also in need of immunological rapid assay technologies to enable the rapid detection of PR residue. The immunoassay techniques exhibit remarkable attributes of exceptional sensitivity, exquisite selectivity, and simplicity, rendering them extensively employed for the detection of residues pertaining to small molecule compounds. Antibody is the core recognition element of immunoassay, which determines the specificity and sensitivity of the method. Currently, there are few studies on immunological rapid assay for PR, and no commercially available enzyme-linked immunosorbent assay (ELISA) kits have been developed. In the existing reports, only Liu et al. obtained specific polyclonal antibodies against PR by immunizing New Zealand white rabbits [17]. The polyclonal antibodies recognize multiple antigenic epitopes [18], which made them susceptible to interference and poorly reproducible. The accuracy and reproducibility of monoclonal antibodies have led to their use in a wide range of medical and testing applications [19]. Therefore, the development of monoclonal antibodies specific to anti-pretilachlor is essential.

The specificity of an antibody determines how widely it can be used in practice. Highly specific monoclonal antibodies are the basis for avoiding false-positive test results [20]. Chloroacetamide herbicides have extremely similar chemical antistructures, which may result in a high cross-reactivity rate of monoclonal antibodies against chloroacetamide drugs, thus it is important to find and adequately expose the key recognition site in hapten design. This study aims at the shortcomings of the rapid detection technology for PR, 3-mercaptopropionic acid was used as a linker arm that coupled PR to bovine serum albumin for the preparation of a more sensitive and specific monoclonal antibody to PR. Based on this antibody, an ic-ELISA method was established for the detection of PR in lake water, soil, and rice. Meanwhile, we employed computer simulation and experimental validation to elucidate the key recognition sites of chloroacetamides. Additionally, we proposed a strategy for enhancing the sensitivity of monoclonal antibodies against PR, thereby providing valuable insights for the development of highly sensitive and specific monoclonal antibodies.

## 2. Materials and Methods

### 2.1. Chemicals and Materials

Alachlor and 3-mercaptopropionic acid were purchased from Macklin (Shanghai, China). Pretilachlor, propisochlor, acetochlor, and metalaxyl were provided from TMRM (Beijing, China) and butachlor was purchased from Aladdin (Shanghai, China). PEG1450, Freund’s complete adjuvant (FCA), hypoxanthine-aminopterin-thymidine (HAT), Freund’s incomplete adjuvant (FIA), Ovalbumin (OVA), Bovine serum albumin (BSA), Peroxidase labeled goat anti-mouse immunoglobulin (HRP-IgG), and 1-Ethyl-3(3-dimethylaminopropyl) Carbodiimide (EDC) were obtained from Sigma (St. Louis, MO, USA), fetal bovine serum were provided from ExCell Bio (Beijing, China). Cell culture plates were obtained from NEST Biotechnology Co., Ltd. (Wuxi, China), N-hydroxysuccinimide (NHS), and other organic chemical reagents were provided from Sinopharm (Shanghai, China).

UV spectrophotometer (UV2600) and high-performance liquid chromatography (LC-20AD) from Shimadzu, Kyoto, Japan. The multifunctional enzyme labeling instrument was from PerkinElmer (Waltham, MA, USA).

All experimental animals used in this study were approved by the Animal Ethics Committee. Female Balb/C mice (6~8 weeks old) were provided by the Laboratory Animal Center of Huazhong Agricultural University.

### 2.2. Synthesis and Computer Simulation of the Hapten PR-SC

The structure of the haptens was plotted by Gauss View 5.0 and then the lowest energy conformation was calculated in Gaussian 09W using density function (DFT), and finally the electrostatic potential (ESP) prediction maps were plotted by Gauss View 5.0.

The synthesis of the hapten PR-SC was based on the method of Liu et al. [17], with some modifications. A total of 2 g 3-mercaptopropionic acid and 15 mL ethanol were added to a flask, followed by 3 g of PR and 0.5 g of potassium hydroxide, and stirred at reflux for 6 h at 70–75 °C. The reaction solution was concentrated under reduced pressure to pale yellow liquid. The liquid was mixed with 20 mL of deionized water and the pH was adjusted to 11 with 1 mol/L NaOH, then centrifuged at 8000 r/min for 5 min and the supernatant was collected. The supernatant was adjusted to pH = 3.0 by 1 mol/L hydrochloric acid and extracted with ethyl acetate, and the organic solvents were combined and concentrated under reduced pressure to obtain the PR hapten (PR-SC) and the molecular weight was identified by IT-TOF. The synthetic route is shown in Figure 1A. 

### 2.3. Preparation of Complete Antigen

Hapten–protein conjugations were prepared by the activated ester method (Figure 1B). Solution A: 80 mg PR-SC were dissolved by 0.3 mL N, N-dimethylformamide in a brown bottle, followed by the sequential addition of 60 mg EDC and 36 mg NHS. The mixture was thoroughly mixed and stirred at room temperature under light exclusion for 12 h. Solution B: 52 mg BSA was dissolved in 5 mL PBS. Solution A was slowly added to solution B in an ice bath, stirred for 12 h at room temperature, and then packed into dialysis bags and dialyzed in PBS for 4 days at 4 °C, and the immunogen (PR-SC-BSA) was obtained. The dialysate was changed three times daily. After dialysis, the reaction solution was centrifuged at 10,000 r/min for 10 min, the resulting precipitate was discarded, and the supernatant was subsequently stored at a temperature of −20 °C. The synthesis of the coating antigen (PR-SC-OVA) followed the same procedure as that for immunogens, with the exception that OVA was used instead of BSA. UV absorption spectroscopy in the range of 200 nm to 400 nm was employed to confirm the successful conjugation of the hapten to the protein.

### 2.4. Production of the Monoclonal Antibody (mAb)

The female Balb/C mice (6~8 weeks old) were immunized with an emulsion of complete antigen PR-SC-BSA and Freund’s adjuvant, administered through multiple subcutaneous injections into the back and neck. FCA was used in the first immunization, and FIA was used in subsequent booster immunizations. A booster vaccination was administered every two weeks for 21 days after the initial immunization, and blood was collected from mice 6–8 days after the final immunization. Ic-ELISA was used to determine the titer and sensitivity of antibodies to PR in mice serum. Cell fusion and ascites production were based on the method previously described [21]. Simply, the mice with the serum showing the best titers were euthanized and SP2/0 myeloma cells were mixed with splenocytes in 50% polyethylene glycol 1450 for 1 min, left to stand for 1 min, and terminated by 40 mL of PBS.

Seven days after cell fusion, positive hybridoma cells were screened by ic-ELISA and subcloned five times to obtain a monoclonal cell line capable of stably secreting antibodies. Selected hybridoma cells were injected into the peritoneal cavity of mice. After 10 days, the mice were euthanized and ascites were collected. The ascites supernatant was stored at −20 °C.

### 2.5. Ic-ELISA Procedure

The ic-ELISA procedure was based on the method of Wu et al. [22], with some modifications. PR-SC-OVA (100 μL, 0.25 μg/mL) was added in microplates and incubated overnight at 4 °C. After being washed by PBST (PBS containing 0.05% Tween 20, pH = 7.4), 1% OVA dissolved in PBS was added to the microplates (200 μL/well) and incubated at 37 °C for 1.5 h. Following three additional washes, 50 μL of PBS or sample solution was added to each microplate, and 8000-fold diluted ascites were added to the microplates in a volume of 50 μL per well, and the reaction was incubated at 37 °C for 40 min. After being washed again, HRP-IgG diluted 10,000-fold was added to each microplate in a volume of 100 μL, and the reaction was incubated at 37 °C for 40 min, protected from light. After being washed three times, TMB substrate solution (100 μL/well) was added and incubated for 15 min at 37 °C away from light. A total of 50 μL H_2_SO_4_ (1 mol/L) was used to terminate the reaction. Finally, the OD450 value was measured by a multifunctional enzyme labeling instrument.

### 2.6. Optimization of Ic-ELISA

PR was used as a competitor and the ic-ELISA was optimized based on previous studies [21], with some modifications. To improve the sensitivity of the reaction, the effects of pH and PR dilutions on the ic-ELISA method were investigated. PBS of different pH (pH = 9.6, 7.4, 5.0) as reaction conditions were used to explore the effect on sensitivity. In addition, the PR standard was diluted with methanol to the appropriate concentration (100 μg/mL), thus PBS with 5, 10%, and 20% methanol were used to check the effect on the sample reconstituted solution as dilutions of PR. The inhibition rate B/B0 (OD of competitor-inhibited wells/OD of blank wells) or the corresponding competitor concentration at 50% inhibition (IC_50_) was used as the assessment of the optimization conditions.

### 2.7. Standard Curve and Cross-Reactivity of Ic-Elisa

PR standard (1 mg/mL in methanol) was diluted to 100, 50, 25, 12.5, 6.25, and 0 μg/L by PBS, and the standard curve was plotted under optimized conditions. To ensure the precision of the standard curve, each set of standard curves was made five times in parallel and repeated for five sets, and the coefficient of variation was used to evaluate the precision of the standard curve. Cross-reactivity (CR) study was performed to determine the specificity of mAb. Five structurally similar pesticides, including alachlor, acetochlor, butachlor, propisochlor, and metalaxyl, were determined by ic-ELISA. CR = (IC_50_ of PR)/(IC_50_ of analyte) × 100%.

### 2.8. Sample Preparation

The rice, lake water, and soil were obtained from local farmers’ markets, lake, and fields, respectively. A total of 20 mL of lake water was taken in a 50 mL centrifuge tube and centrifuged at 8000 r/min for 10 min, 10 mL of supernatant was taken in a 50 mL centrifuge tube and 10 mL of sample diluent (PBS with 5% methanol) was added, mixed thoroughly and determined.

Rice and soil: 1 g of sample was weighed in a 50 mL centrifuge tube, 10 mL of PBS solution containing 5% methanol was added, the mixture was vibrated for 5 min and then left to stand for 20 min, then centrifuged at 8000 r/min for 10 min, and the supernatant was taken for the determination.

### 2.9. Validation of the Ic-ELISA

The validation of the ic-ELISA was performed by detecting blank samples, sample spiking recovery, and comparison of HPLC assay results. Twenty blank samples of soil, lake water and rice (PR was not detected by HPLC) were taken separately and pretreated according to the sample treatment method described above and then performed to ic-ELISA. LOD and LOQ were calculated according to the following equations: LOD = C + 3 × SD, LOQ = C + 10 × SD (C: Average concentration of target molecule detected in 20 blank samples; SD: Standard deviation of target molecule concentrations detected in 20 blank samples).

The accuracy and precision of the ic-ELISA method were determined by analyzing the spiked recoveries of the samples. Three different concentrations of PR (10, 20, and 40 μg/kg) were spiked into blank lake water, soil, and rice samples, and the recoveries were calculated by the following formula: Recovery (%) = Sample Detection Concentration/Actual Spiked Concentration × 100%.

The reliability of the ic-ELISA method was confirmed in this study through its validation using HPLC. The HPLC method employed was performed the method of Luo et al., with appropriate modifications [23]. Lake water was assayed by ic-ELISA, and HPLC results were used as reference standards. Lake water sample was spiked with PR solutions at concentrations of 50, 100, 200, 400, and 800 μg/L, each divided into two parts, and processed as follows: 100 mL of lake water samples were measured accurately and placed in a 250 mL dispensing funnel. A total of 10 g of sodium chloride and 3 drops of glacial acetic acid were added and the solids were shaken to dissolve. The sample was extracted three times with 30 mL, 30 mL, and 20 mL of dichloromethane, and the combined organic solvents were dried and concentrated on a rotary evaporator. The concentrated samples were re-dissolved and assayed by ic-ELISA and HPLC, respectively.

Chromatographic column: C18 (250 mm × 4.6 mm × 5.0 μm); Mobile phase: methanol-water (75:25, *v*/*v*); Flow rate: 1 mL/min; Detection wavelength: 220 nm; Column temperature: 40 °C; Injection volume: 20 μL.

## 3. Results

### 3.1. Identification of Hapten

The design of haptens should aim to closely mimic the target molecule in terms of morphology, structure, and electron distribution. This approach facilitates the preparation of monoclonal antibodies with enhanced sensitivity and specificity. PR is a chloroacetamide herbicide with a structure consisting of three groups: a benzene ring, an N-chloroacetamido group, and a hydrocarbon-oxygen chain. The hydrocarbon-oxygen branched chain is the main group that distinguishes amide herbicides, therefore the modification of the hapten should be carried out from the benzene ring and the N-chloroacetamide group. Modification of the benzene ring requires addressing two issues: protection of the benzene ring branched chain and the immunogenicity of the hapten. The addition of branched chains to the benzene ring can damage the characterization of other branched chains, which greatly limits the modification of the benzene ring. The benzene ring has been reported to affect the immunogenicity of antigens [24], therefore modifications to benzene rings should be avoided. The N-chloroacetamide moiety was chosen for the synthesis of PR hapten (PR-SC), which allows adequate exposure of the benzene ring and hydrocarbon-oxygen branched chain, thereby improving the immunogenicity and specificity of the hapten and minimizing cross-reactivity with similar herbicides.

The Mulliken charges of PR and hapten PR-SC were calculated and analyzed from 1C to 21C. As shown in Figure 2, the Mulliken charges of the remaining atoms of hapten PR-SC and PR maintain a high degree of consistency with each other, except for the change of charge after the substitution of 13Cl by S. Particularly, the Mulliken charges of 14C, 15C, 16O, 17C, 20C, and 21C constituting the hydrocarbon-oxygen branched chain maintain perfect consistency. This allows the specificity of the antibody for PR to be fully assured.

The molecular formula of PR is C_17_H_26_ClNO_2_, while the molecular formula of PR-SC is C_20_H_31_NO_4_S. The PR-SC synthesis employs 3-mercapto propionic acid to substitute the chlorine atoms on the PR, resulting in carboxyl groups. As shown in Appendix A, PR-SC was successfully synthesized with IT-TOF verification, the theoretical value of [M + Na]^+^ for PR-SC was 404.1866 and the measured value was 404.1868. The theoretical value of [M + H]^+^ for PR-SC was 382.2047 and the measured value 382.2082. The high consistency between the observed and predicted values of [M + Na]^+^ and [M + H]^+^ for PR-SC indicates the accurate determination of its molecular weight and successful synthesis. 

### 3.2. Identification of Complete Antigens

The UV absorption spectra of the hapten PR-SC exhibited maximum peaks at 228 nm, while those of BSA and OVA were observed at 279 nm in Figure 3. Notably, the UV absorption peaks of the conjugates PR-SC-BSA and PR-SC-OVA displayed significant shifts compared to those of PR-SC, BSA, and OVA, providing evidence for the successful synthesis of complete antigens. Moreover, efficient coupling between PR-SC and vector proteins was achieved with coupling ratios of 15.4:1 (PR-SC-BSA) and 10.2:1 (PR-SC-OVA).

### 3.3. Production and Identification of the mAb

The selection of mouse PRA2 for cell fusion was based on the antiserum results obtained from four immunized mice, as presented in Appendix A. The PR/114 cell line was subsequently screened from the fused cells through subcloning and ic-ELISA (PR concentration: 50 μg/L, inhibition rate: 42%, see Appendix A). Antibody ascites were prepared using PR/114, and the subtype of this monoclonal antibody was identified as IgG1 through kit analysis. The concentration of antibody ascites obtained was determined to be 30.3 mg/mL.

### 3.4. Optimization of Ic-ELISA Conditions

#### 3.4.1. Optimization of Dilution of mAb and Coating Antigen

MAb and coating antigen at optimal dilution concentration, OD value at 450 nm was 2.0. As shown in Appendix A, there were three combinations of coating antigen dilutions and mAb: (0.25 μg/mL, 0.5 μg/mL), (0.125 μg/mL, 0.5 μg/mL), and (0.0625 μg/mL, 1 μg/mL). The IC_50_ values of the three combinations were calculated by ic-ELISA, which were 47.06 μg/L, 35.41 μg/L, and 31.47 μg/L, respectively (Appendix A), which led to the conclusion that the third combination was the optimal dilution concentration.

#### 3.4.2. Optimization of Drug Dilution Conditions

The ic-ELISA was conducted at various pH levels to investigate the impact of pH on mAb sensitivity. The results are presented in Figure 4A, revealing that the IC_50_ value is minimized at pH = 7.4. Consequently, PBS with a pH of 7.4 was selected as the optimal dilution solution for both the antibody and drug.

The PR standard master mix was prepared in methanol, and it was imperative to investigate the potential impact of methanol on the antibody. Ic-ELISA was performed with PBS solutions containing 0%, 5%, 10%, and 20% methanol as dilutions for PR standards. The results are shown in Figure 4B. The IC_50_ was minimized when 5% methanol PBS solution was used as the drug diluent, thus establishing it as the preferred standard diluent. It is worth mentioning that PR exhibits solubility in methanol but has limited solubility in water. The addition of an appropriate amount of methanol to the diluent improves the solubility of PR, which increases the probability that the antibody will bind to the drug, which could explain the ability of methanol to improve the recognition of PR by the PR/114 antibody.

### 3.5. Evaluation of the Optimized Ic-ELISA

#### 3.5.1. Ic-ELISA Calibration Curve

The standard solution was diluted to concentrations of 6.25 μg/L, 12.5 μg/L, 25 μg/L, 50 μg/L, and 100 μg/L to establish the PR standard curve under optimized ic-ELISA conditions. Using the logarithm of PR concentration as the *X*-axis and B/B0 as the *Y*-axis, the regression equation of PR/114 antibody ic-ELISA was Y = 1.08288 − 0.387X, R^2^ = 0.9915, and the IC_50_ value was 31.47 ± 2.35 μg/L with a linear range of 6.25~100 μg/L (Figure 5). Appendix A showed that both intra-plate coefficient of variation and inter-plate coefficient of variation were less than or equal to15.72% and7.79%, respectively, indicating favorable precision for this established standard curve, while also demonstrating greater sensitivity compared with antibodies obtained by Liu et al. (IC_50_ = 35.9 μg/L) [17].

#### 3.5.2. Cross-Reactivity

The five chloroacetamide herbicides, namely alachlor, acetochlor, butachlor, propisochlor, and metalaxyl were formulated at appropriate concentrations. Standard curves for these compounds were established and the IC_50_ values were separately calculated using optimized ic-ELISA conditions. The IC_50_ values of the PR were used as the standard to compare and calculate the cross-reactivity rates of other compounds. The results are shown in Table 1. The cross-reactivity rates of the antibody against these five chloroacetamide herbicides were all less than 3.0%.

The cross-reactivity of the PR/114 antibody with alachlor, acetochlor, metalaxyl, butachlor, and propisochlor were all less than 3.0%. As depicted in Table 1 alachlor, acetochlor, butachlor, and propisochlor exhibit distinct hydrocarbon-oxygen branched chains: Alachlor and acetochlor lack a methyl group compared to PR; Butachlor possesses an additional methyl group relative to PR; while propisochor differs in its linkage to the propyl group of PR. The difference in hydrocarbon-oxygen branched chains has a strong influence on the recognition ability, which leads to the assumption that the hydrocarbon-oxygen branched chain is the key recognition site.

Computer simulations of five structurally similar drug structures were performed as described previously to obtain minimum energy conformations, ESP values, and Mulliken charges data to analyze the cross-reactivity of the antibodies. The lowest energy conformational maps and ESP distributions of PR and five structurally similar drugs based on Gaussian 09W calculations and rendered by GaussView 5.0 were shown in Figure 6A. The ESP of the benzene ring and the N-chloroacetamido group of PR were comparable to those of the other five drugs, while exhibiting varying degrees of ESP distribution in the hydrocarbon-oxygen branched chain. Additionally, Mulliken charges were calculated for PR and its five analogs based on their lowest energy conformations, as depicted in Figure 6B. For simplicity, we disregarded 1C-9C due to their closely similar Mulliken charge values. Notably, significant differences in the Mulliken charges observed at positions 17C, 20C, and 21C (red box in Figure 6B) within the hydrocarbonyloxy branched chains further emphasize their crucial role as specificity determinants. This finding aligns with both ESP distributions and experimental results.

The difference between the coating antigen and target molecules and immunogens is one of the factors to improve antibody sensitivity [25]. Based on the above simulation results, the hydrocarbon-oxygen branch of chloroacetamides has been shown to be a site of specificity for the target molecule, thus, it is hypothesized that the use of other chloroacetamide herbicides similar to PR as heterologous coating antigens may significantly increase antibody sensitivity by increasing the competitiveness of the antibody against the PR molecule. This conclusion is subject to further experimental verification.

### 3.6. Validation of Ic-ELISA Method

#### 3.6.1. Optimization of Sample Pretreatment

For the pretreatment of the lake water sample, a dilution method was used, which was simple and less time-consuming. Considering the more complex water environment that may be encountered in practical application, at least 1-fold dilution is required when processing lake water samples. PR boiling point was 135 °C, and the recovery of PR in the sample was found to be low when extracted with organic solvent and concentrated, which was inferred to volatilize PR when organic solvent evaporated. Therefore, the extraction method for PR in soil and rice should be diluted in PBS containing 5% methanol for extraction.

The OD values of 20 blank samples were measured for lake water, rice, and soil, respectively, and the concentration of PR in the blank samples was calculated based on the standard curve of the PR/114 antibody ic-ELISA to calculate the LOD and LOQ for each sample. The results are shown in Table 2. The limits of detection for PR in lake water, rice, and soil were 2.37~4.83 μg/L and the limits of quantification were 4.46~10.13 μg/L. The average recoveries in the three samples were 78.3~91.3% and the coefficients of variation were less than 9.5%.

#### 3.6.2. Comparison of Ic-ELISA and HPLC

The results of the ic-ELISA and HPLC determination of lake water were detailed in Table 3, with recoveries ranging from 81.0% to 96.2% for both methods. As shown in Figure 7, the correlation between ic-ELISA and HPLC was good and the linear correlation coefficient was R^2^ = 0.9913. The results indicated that the ic-ELISA method established in this study can be used as a reliable tool for the detection of PR in real samples.

## 4. Conclusions

In this study, a monoclonal antibody with enhanced sensitivity and specificity towards PR was prepared with an IC_50_ of 31.47 μg/L, while showing no cross-reactivity with other chloroacetamide herbicides. An ic-ELISA method has been developed for the detection of PR residues in the environment and cereals, with a linear range of 6.25~100 μg/L. The LOD of PR in lake water, rice, and soil samples were 2.37~4.83 μg/L and the LOQ were 4.46~10.13 μg/L, the recoveries were 78.3~91.3% with the coefficients of variation less than 9.5%. The method was confirmed by HPLC, confirming the reliability of the ic-ELISA was reliable. We have provided a more sensitive method of detecting PR in the environment, and a new way of monitoring environmental safety that was scientifically important and market value.

## Figures and Tables

**Figure 1 foods-13-00012-f001:**
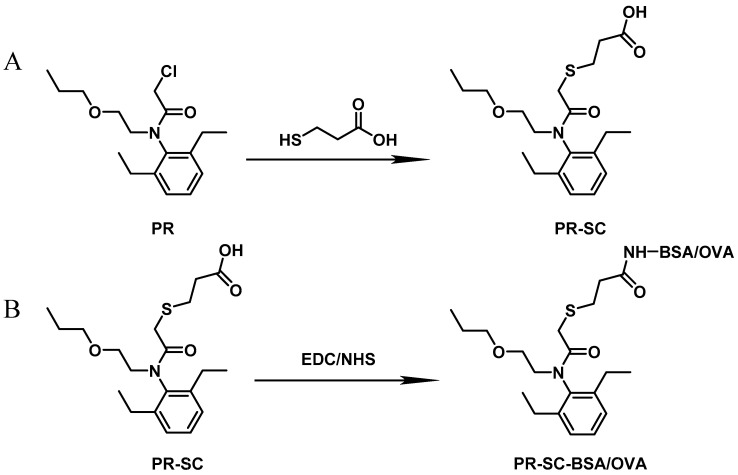
(**A**): Synthesis route of PR hapten (PR-SC) (**B**): Synthesis of antigens of PR-SC-BSA/OVA.

**Figure 2 foods-13-00012-f002:**
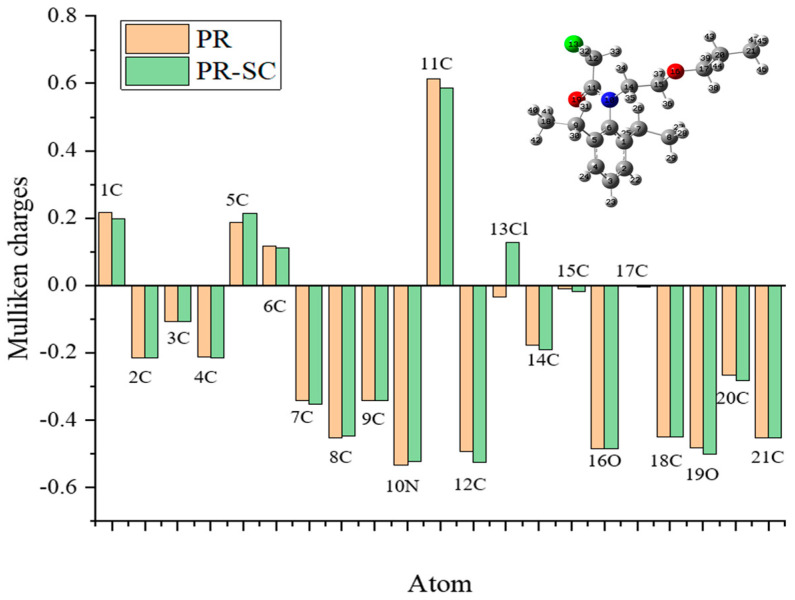
Mulliken charges of PR and hapten PR-SC.

**Figure 3 foods-13-00012-f003:**
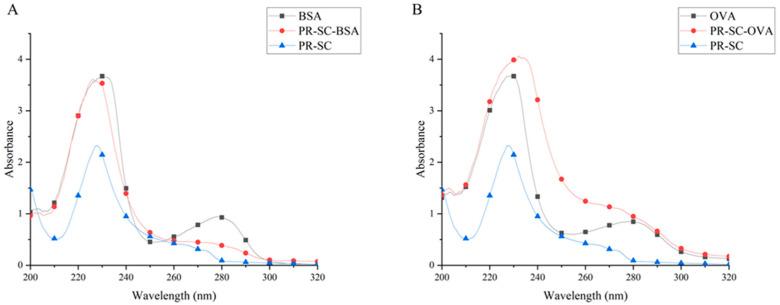
Analysis of PR-SC-BSA (**A**) and PR-SC-OVA (**B**) conjugations by UV–visible spectroscopy.

**Figure 4 foods-13-00012-f004:**
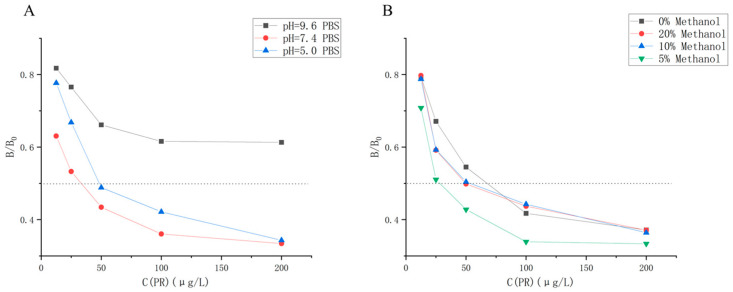
(**A**): Inhibition rate-PR concentration curve at pH = 9.6, 7.4, 5.0 in PBS. (**B**): Inhibition rate-PR concentration curve of PBS containing 0%, 5%, 10%, and 20% Methanol as a drug diluent.

**Figure 5 foods-13-00012-f005:**
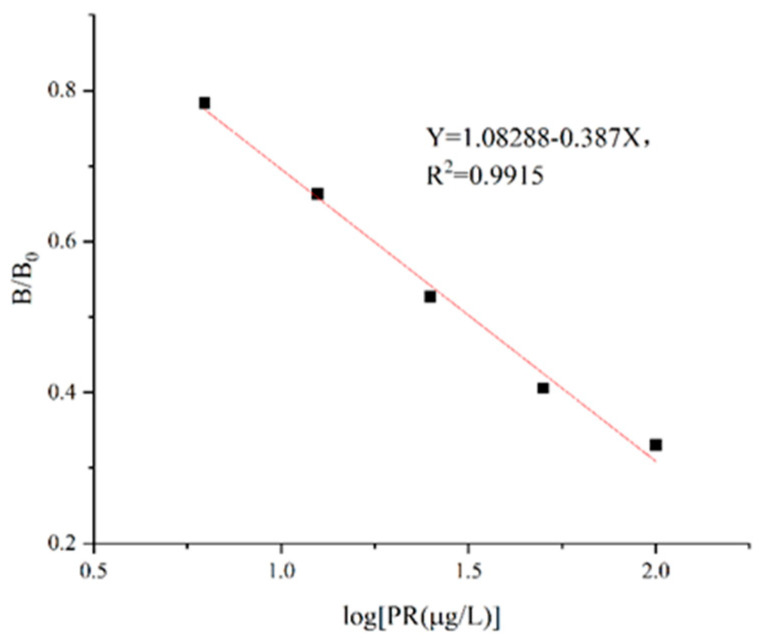
Standard curve of ic-ELISA for PR/114 antibody.

**Figure 6 foods-13-00012-f006:**
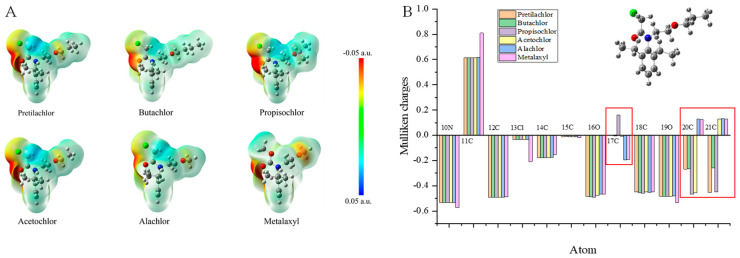
(**A**): Surface ESP distribution of PR and its analogs. (**B**): Mulliken charges of PR and its analogs.

**Figure 7 foods-13-00012-f007:**
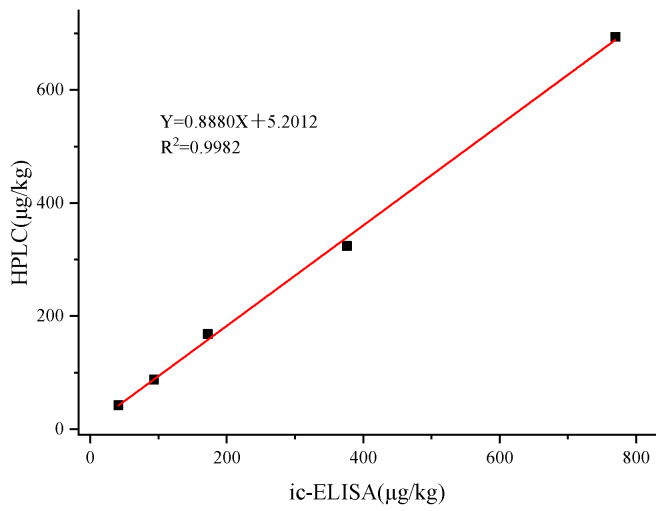
Comparison of ic-ELISA and HPLC for the detection of PR (Lake water).

**Table 1 foods-13-00012-t001:** Cross-reactivity of PR/114 antibody to chloroacetamide herbicides.

Chloroacetamide Herbicides	Structure	IC_50_ (μg/L)	Cross-Reactivity (%)
PR	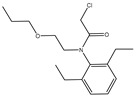	31.47	100
Alachlor	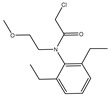	>3000	<1
Acetochlor	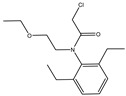	>3000	<1
Propisochlor	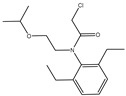	1541	2.0
Butachlor	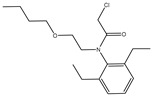	>3000	<1
Metalaxyl	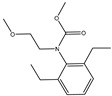	1927	1.6

**Table 2 foods-13-00012-t002:** The recovery rate of PR in three samples.

Samples	LOD (μg/L)	LOQ (μg/L)	Spiked (µg/L)	Recovery (C ± SD, %, *n* = 15)	Coefficient of Variation (CV) (%, *n* = 15)
Lake water	4.83	10.13	10.00	89.7 ± 8.3	9.3
			20.00	85.9 ± 7.0	8.1
			40.00	83.9 ± 8.0	9.5
Rice	3.04	5.68	10.00	81.3 ± 2.5	3.1
			20.00	91.3 ± 3.0	3.3
			40.00	78.3 ± 2.3	2.9
Soil	2.37	4.46	10.00	86.5 ± 4.2	4.8
			20.00	84.4 ± 2.3	2.7
			40.00	86.6 ± 4.3	5.0

**Table 3 foods-13-00012-t003:** Results of ic-ELISA and HPLC assay (Lake water).

Spiked (μg/kg)	ic-ELISA (*n* = 5, μg/kg)	ic-ELISA Recovery (%)	HPLC (*n* = 5, μg/kg)	HPLC Recovery (%)
50	41.22	82.4	42.39	84.8
100	93.45	93.5	87.48	87.5
200	172.33	86.2	168.43	84.2
400	376.22	94.1	323.92	81.0
800	769.53	96.2	693.82	86.7

## Data Availability

The data presented in this study are available on request from the corresponding author.

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
