# Peer review of "A Sensitive and Specific Monoclonal Antibody Based Enzyme-Linked Immunosorbent Assay for the Rapid Detection of Pretilachlor in Grains and the Environment"

_foods, 2023, doi:10.3390/foods13010012_

Round 1

Reviewer 1 Report

Comments and Suggestions for Authors

The Manuscript is well-written and organized.  The following facts should be improved and corrected:

The ic-ELISA method was validated by high-performance liquid chromatography, thus, it can be used for residue detection of pretilachlor in the environment and grains- The ELISA method was not validated by HPLC, HPLC is just used as a confirmatory method. Please correct this.

PR has low toxicity, but you also wrote that it may cause cancer in humans, the toxic effects include the emergence of neurotoxicity, genotoxicity, and carcinogenicity. Please correct this.

The main methods for residue analysis of PR in domestic and overseas are gas chromatography and liquid chromatography- the main method, or the most commonly used method? Please write the most commonly used detectors for gas chromatography and liquid chromatography.

Please write a reference for this sentence, because some of these claims are disputable because these methods have also a lot of advantages in comparison to other and ELISA methods: Despite these instrument detection methods having the advantages of high sensitivity and accuracy, they suffer from disadvantages of complicated operation, long time consumption, high detection cost, high use of organic solvents, and high requirements for instruments and operators.

The ic-ELISA validation was performed as previously reported [23]. – In accordance with which official procedure and/or recommendation validation procedure was performed? Which parameters were determined?

Twenty different blank samples- which kind of sample, rice, lake water, or soil? Each matrix must be separately validated.

The limit of detection (LOD) and limit of quantification (LOQ) were calculated by the average drug concentration detected in the samples (C) and the standard deviation (SD) of the measured drug concentration.- DRUG concentration?

In the text, You wrote that the HPLC method was used as a comparative method, while in Table 3 You wrote LC-MS/MS. Please write the whole name of the method LC-MS/MS, and use this abbreviation in the text instead of HPLC.

For analysis of food contaminants, such as pretilachlor, there is a need to use certified reference material for validation. This is the best way to investigate the accuracy of the newly developed method. 

Comments on the Quality of English Language

Minor revision

Author Response

Response to Reviewer #1:

The Manuscript is well-written and organized.  The following facts should be improved and corrected

Response: Thank you so much for your encouraging and helpful comments on our manuscript. Your support truly motivates us to keep pushing forward. The suggestions and issues you have raised are incredibly valuable to the article's improvement. We have carefully revised it line by line, highlighting the changes in red for your convenience during review. Once again, we sincerely appreciate your time and effort.

  1. The ic-ELISA method was validated by high-performance liquid chromatography, thus, it can be used for residue detection of pretilachlor in the environment and grains- The ELISA method was not validated by HPLC, HPLC is just used as a confirmatory method. Please correct this.

Response: Thank you so much for bringing to our attention the areas where our manuscript can be improved. Your suggestion that HPLC is only used as a confirmatory method and not to validate ELISA methods is very rigorous and correct! Based on your suggestions, We have replaced the original sentence with the following description and marked it in red to make it easier for your review, lines 26-28: The reliability of the ic-ELISA method for residue detection of pretilachlor in the environment and grains was confirmed using high performance liquid chromatography.

  1. PR has low toxicity, but you also wrote that it may cause cancer in humans, the toxic effects include the emergence of neurotoxicity, genotoxicity, and carcinogenicity. Please correct this.

Response: Thank you so much for pointing out the shortcomings of our manuscript. The advice you provided has highlighted a flaw in the logic of our manuscript, and we sincerely appreciate your thorough and meticulous review! PR is a low toxicity pesticide compared to most highly toxic pesticides, we have recorrected the description of lines 38-43 of the manuscript as follows: But its overuse has potential problems, such as increased weed resistance and contamination of the surrounding environment, PR has also shown toxic effects on fish[2, 4-7], and algae[8-9]. It is important to note that the perceived low toxicity of PR is relative and prolonged ultra-limited exposure still carries risks. Some studies have shown PR to be neurotoxic, genotoxic and carcinogenic[10], its enrichment in the food chain ultimately harms human beings[11-12].

  1. The main methods for residue analysis of PR in domestic and overseas are gas chromatography and liquid chromatography- the main method, or the most commonly used method? Please write the most commonly used detectors for gas chromatography and liquid chromatography.

Response: Thank you so much for bringing to our attention the areas where our manuscript can be improved. Gas chromatography and liquid chromatography are the most commonly used instrumental methods for detecting PR, and in response to your suggestion, we have provided a brief description of the instrument's detector in the red-highlighted section of lines 47-56 of the manuscript. Your suggestion has enriched the content of the manuscript, thank you again!

  1. Please write a reference for this sentence, because some of these claims are disputable because these methods have also a lot of advantages in comparison to other and ELISA methods: Despite these instrument detection methods having the advantages of high sensitivity and accuracy, they suffer from disadvantages of complicated operation, long time consumption, high detection cost, high use of organic solvents, and high requirements for instruments and operators.

Response: Thank you very much for pointing out the shortcomings of our manuscript. This suggestion pinpoints very accurately the controversial descriptions present in our manuscripts. Based on your suggestions, we have revised this section to the following description: These instrumental assays offer the advantage of high sensitivity and accuracy, however, the market is also in need of immunological rapid assay technologies to enable the rapid detection of PR residue. This section is redlined for your review in lines 56-58 of the manuscript.

  1. The ic-ELISA validation was performed as previously reported [23]. – In accordance with which official procedure and/or recommendation validation procedure was performed? Which parameters were determined?

Response: Thank you very much for pointing out the shortcomings of our manuscript. We made an error in the description and citation of references in this section, and once again your suggestion accurately points out the issues with the manuscript. The ic-ELISA validation was performed as previously reported [23]. We sincerely apologize for the confusion. The description here does not match the inserted reference, we have replaced the original text with the following description: The validation of the ic-ELISA was performed by detecting blank samples, sample spiking recovery, and comparison of HPLC assay results. The reference cited here is the HPLC assay for PR, which we are using for comparison with the ic-ELISA method. Based on the above explanation, we have reworked lines 196-203 of the manuscript and cited the reference in the correct place (line 210). Thanks again for your precise suggestions!

  1. Twenty different blank samples- which kind of sample, rice, lake water, or soil? Each matrix must be separately validated.

Response: Thank you very much for pointing out the shortcomings of our manuscript. Your comment made us realize that the description of this part of the manuscript was too vague. The 20 blank samples in the manuscript refer to 20 blank samples for each of lake water, soil, and rice, not just one kind, and the LOD and LOQ in Table 2 demonstrate that we validated each of the blank samples. The description of this section has been revised in line 197-198 of the manuscript, and we would like to ask you to review it again.

  1. The limit of detection (LOD) and limit of quantification (LOQ) were calculated by the average drug concentration detected in the samples (C) and the standard deviation (SD) of the measured drug concentration.- DRUG concentration?

Response: Thank you very much for pointing out the shortcomings of our manuscript. We apologize for the confusion caused by the description in the manuscript. Because ELISA has background interference, even a blank sample will detect a value that is so small as to be negligible. The drug concentration here refers to the concentration of PR detected in the blank sample. We have redescribed C and SD in lines 201-203 of the manuscript and changed the drug concentration to the target molecule concentration to avoid causing confusion to the reader. Changes to this part of the manuscript were described below: LOD and LOQ were calculated according to the following equations: LOD=C+3×SD, LOQ=C+10×SD (C: Average concentration of target molecule detected in 20 blank samples; SD: Standard deviation of target molecule concentrations detected in 20 blank samples).

  1. In the text, You wrote that the HPLC method was used as a comparative method, while in Table 3 You wrote LC-MS/MS. Please write the whole name of the method LC-MS/MS, and use this abbreviation in the text instead of HPLC.

Response: Thank you very much for pointing out the shortcomings of our manuscript. We appreciate that you have pointed out this error in our manuscript. The instrumental comparison method we used was HPLC, not LC-MS/MS, and the description of Table 3 was incorrect, which we have corrected in lines 376 and 381 of the manuscript.

  1. For analysis of food contaminants, such as pretilachlor, there is a need to use certified reference material for validation. This is the best way to investigate the accuracy of the newly developed method.

Response: Thank you very much for pointing out the shortcomings of our manuscript. You are absolutely and completely right to make this suggestion! The HPLC method we referenced was an article published in the Journal of Administration and Technique of Environmental Monitoring, and was not a method we arbitrarily used. Another factor in choosing the HPLC method described in this article is that the samples used in this method are also water, soil, and rice, which are identical to the samples used in this study. The reference has been corrected on line 211 of the manuscript. Thank you once again for your meticulous review, which greatly enhances the professionalism and academic rigor of our paper!

Reviewer 2 Report

Comments and Suggestions for Authors

The submitted manuscript reports a study of a monoclonal antibody based ELISA for the rapid detection of pretilachlor molecule.

The experimental design is sufficiently detailed and fundamentally correct.

I think the authors need to improve the quality of the manuscript:

- English needs to be profoundly revised (specific comments reported in the next section)

- the comparisons of ic-ELISA and HPLC in the validation phase are reported, but it is not clear whether chromatography is coupled to a mass spectrometer (HPLC MS/MS) for the validation as reported in the caption of table 3 , or coupled to UV-vis detector as reported in section 2.9 of the Materials and Methods paragraph

- the comparison of the results of ic-ELISA and LC-MS/MS assay is reported only for lake water. The authors should similarly show the results for soil and rice

I therefore suggest a major revision to the authors

Comments on the Quality of English Language

English needs a strong revision.

Verbs are often missing and sometimes the language is more similar to a laboratory notebook than to a description for a scientific publication

Author Response

Response to Reviewer #2:

The submitted manuscript reports a study of a monoclonal antibody based ELISA for the rapid detection of pretilachlor molecule.

The experimental design is sufficiently detailed and fundamentally correct.

Response: Thank you so much for your encouraging and helpful comments on our manuscript. Your support truly motivates us to keep pushing forward. The suggestions and issues you have raised are incredibly valuable to the article's improvement. We have carefully revised it line by line, highlighting the changes in red for your convenience during review. Once again, we sincerely appreciate your time and effort.

  1. - English needs to be profoundly revised (specific comments reported in the next section)

Response: Thank you very much for pointing out the shortcomings in our manuscript! We have corrected the English expression of the whole manuscript, we hope that you will be able to review it once again.

  1. - the comparisons of ic-ELISA and HPLC in the validation phase are reported, but it is not clear whether chromatography is coupled to a mass spectrometer (HPLC MS/MS) for the validation as reported in the caption of table 3, or coupled to UV-vis detector as reported in section 2.9 of the Materials and Methods paragraph.

Response: Thank you so much for bringing to our attention the shortcomings of our manuscript. We truly appreciate your valuable feedback in identifying this error in our manuscript. The analytical method we employed was HPLC, rather than LC-MS/MS, and the description of Table 3 was inaccurate, which has been rectified in lines 376 and 381 of the manuscript.

  1. - the comparison of the results of ic-ELISA and LC-MS/MS assay is reported only for lake water. The authors should similarly show the results for soil and rice.

Response: Thank you so much for bringing to our attention the shortcomings of our manuscript. We apologize for not being able to include this information in the manuscript as we lack of control trial data for soil and rice samples. Considering the consistent recoveries observed for both HPLC and our established ic-ELISA method across diverse sample types, we believe that the reliability of the ic-ELISA method can be validated by conducting a comparative analysis on a single representative sample. In addition, we reviewed other similar literature, and most of the literature also used a single sample as a control experiment to verify the reliability of the ELISA method when comparing the instrumental method with ELISA. The relevant literature has been compiled for your reference.

Ren, Y.; Tian, R.; Wang, T.; Cao, J.; Li, J.; Deng, A. An Extremely Highly Sensitive ELISA in pg mL−1 Level Based on a Newly Produced Monoclonal Antibody for the Detection of Ochratoxin A in Food Samples. Molecules 202328, 5743. https://doi.org/10.3390/molecules28155743

Huang, L., Chen, H., Cui, P., Ding, Y., Wang, M., & Hua, X. (2022). Development of immunoassay based on rational hapten design for sensitive detection of pendimethalin in environment. Sci Total Environ, 830, 154690. https://doi.org/10.1016/j.scitotenv.2022.154690.

Wu, S., Li, H., Yin, X., Si, Y., Qin, L., Yang, H., . . . Peng, D. (2022). Preparation of Monoclonal Antibody against Pyrene and Benzo [a]pyrene and Development of Enzyme-Linked Immunosorbent Assay for Fish, Shrimp and Crab Samples. Foods, 11(20). https://doi.org/10.3390/foods11203220.

  1. English needs a strong revision. Verbs are often missing and sometimes the language is more similar to a laboratory notebook than to a description for a scientific publication.

Response: Thank you for your valuable suggestions regarding the language used in the manuscript. We have made modifications to the expression throughout the entire manuscript and corrected grammatical errors that were present. We hope that you will be able to review it once again.

Round 2

Reviewer 2 Report

Comments and Suggestions for Authors

Authors followed the suggestions and modified the manuscript accordingly.

Even if I do not fully agree with the response to my comment #3, I believe that the manuscript could be published consistent with the comments of other reviewers.